# Nanotechnology-Based Delivery Systems for Antimicrobial Peptides

**DOI:** 10.3390/pharmaceutics13111795

**Published:** 2021-10-26

**Authors:** Adewale Oluwaseun Fadaka, Nicole Remaliah Samantha Sibuyi, Abram Madimabe Madiehe, Mervin Meyer

**Affiliations:** Department of Science and Innovation (DSI)/Mintek Nanotechnology Innovation Centre (NIC), Biolabels Node, Department of Biotechnology, University of the Western Cape, Bellville 7535, South Africa

**Keywords:** antimicrobial peptides, antimicrobial resistance, nanotechnology, nanocarriers, drug resistance, nanoparticles, drug delivery systems

## Abstract

Antimicrobial resistance (AMR) is a significant threat to global health. The conventional antibiotic pool has been depleted, forcing the investigation of novel and alternative antimicrobial strategies. Antimicrobial peptides (AMPs) have shown potential as alternative diagnostic and therapeutic agents in biomedical applications. To date, over 3000 AMPs have been identified, but only a fraction of these have been approved for clinical trials. Their clinical applications are limited to topical application due to their systemic toxicity, susceptibility to protease degradation, short half-life, and rapid renal clearance. To circumvent these challenges and improve AMP’s efficacy, different approaches such as peptide chemical modifications and the development of AMP delivery systems have been employed. Nanomaterials have been shown to improve the activity of antimicrobial drugs by providing support and synergistic effect against pathogenic microbes. This paper describes the role of nanotechnology in the targeted delivery of AMPs, and some of the nano-based delivery strategies for AMPs are discussed with a clear focus on metallic nanoparticle (MNP) formulations.

## 1. Introduction

Antibiotics have been used for decades to cure infectious diseases and enabled most of modern medicine. Without antibiotics, even routine medical procedures can lead to life-threatening infections. The first widely used antibiotic, penicillin, was discovered in 1928. In 1945, Alexander Fleming warned that bacterial resistance had the potential to ruin the miracle of antibiotics [1]. Shortly thereafter, beta-lactam antibiotics were discovered and proved to be effective against penicillin-resistant microbes [2,3]. This was followed by the methicillin-resistant *Staphylococcus aureus* (MRSA) [4]; since then, resistance has been reported against almost all known antibiotics to date [5]. In the high-priority list of antibiotic-resistant strains that require urgent attention are *Enterococcus faecium*, *Staphylococcus aureus* (*S. aureus*), *Klebsiella pneumoniae (K. pneumoniae*), *Acinetobacter baumannii*, *Pseudomonas aeruginosa* (*P. aeruginosa*), and *Enterobacter species* (ESKAPE). The ESKAPE are among the 12 microorganisms that are listed as critical to medium priority pathogens by the World Health Organization (WHO). These bacteria place a significant burden on the healthcare systems and global economic costs [2,3]. Efforts to impede the spread of these pathogens have been hindered by their ability to resist antibacterial drugs.

As it stands, the misuse and overuse of these drugs are the major contributing factors toward antibiotic resistance, which often reduces the efficacy of newly discovered antibiotics and their derivatives [6]. Antibiotic-resistant infections account for about 700,000 mortalities per year, which is estimated to increase to over 10 million deaths by 2050, making AMR a global health crisis [7]. AMR occurs naturally, but the process has been accelerated by the misuse of antibiotics in both humans and animals. As reported by WHO, increased hospitalization, higher medical costs, and elevated mortality have been associated with AMR. AMR threatens the successful treatment of infections caused by drug-resistant microbes [8]. Infectious diseases such as pneumonia, tuberculosis, gonorrhea, malaria, HIV/AIDS, and salmonellosis are becoming increasingly difficult to manage as known potent antimicrobial agents are becoming less effective. In this context, the loss of antimicrobial potency due to resistance in addition to the lack of new and alternative antimicrobial agents underscores the requirement for novel therapeutic agents.

AMPs of natural and artificial origin have gained attention in recent years due to their biological activities as potential alternatives to conventional antimicrobial agents that can combat pathogenic and drug-resistant microorganisms [9]. These biomolecules serve as a natural first-line of defense system against invading pathogens, working either individually or synergistically with the innate immune system to inhibit the growth of pathogenic microorganisms. AMPs are divided into four categories based on their structural conformation and characteristics. This class of peptides (AMPs) are stable over a wide pH range, can work in synergy with immune cells, and possess effective antimicrobial activities against all kinds of pathogens including viruses. The discovery of defensins among other identified AMPs have been a major breakthrough as alternative molecules for use against antibiotic-resistance and in the development of novel antimicrobial agents [10].

Over 14 manually curated AMP databases have been developed to provide information and enable researchers to synthesize AMPs with better therapeutic index. Examples of these databases include the Database of Antimicrobial Activity and Structure of Peptides (DBAASP v3.0) [11], The Antimicrobial Peptide Database (APD3) [12], dbAMP [13], Yet Another Database of Antimicrobial Peptides (YADAMP), etc. To give insight into the growing number of available AMPs, the APD3 for example is composed of 3257 AMPs from six kingdoms (bacteria, archaea, protists, fungi, plants, and animals) with a broad spectrum of antimicrobial activities [14].

AMPs have been used in the treatment of microbial infections emerging from bacteria, fungi, and viruses [15]. The two most studied AMPs of human origin are the cathelicidin LL-37 and defensins. The antiviral effect of defensins has been demonstrated against human viral infections [16,17,18]. The recombinant human β-defensins (mouse β-defensin 3) showed antiviral activity against influenza A virus in both in vitro and in vivo studies [17]. Several studies have also demonstrated the immunomodulatory [19,20], antimicrobial [21], and wound-healing activities of the human LL-37 [22]. Of note, LL-37 interacts with keratinocytes through the P2X7–SFK–Akt–CREB/ATF1 signaling [23,24] and COX-2 [25] pathways as well as with fibroblasts through the P2X7R pathway [26] and the protein kinase/ERK pathway, supporting its healing effect in polymicrobial-infected wounds [27,28]. 

Despite the health properties offered by AMPs, they have some limitations, which ultimately delay their progress into clinical trials [15]. Chemical modification of the AMPs and the use of delivery systems have been reported to improve their pharmacokinetics [29]. Nanotechnology-based delivery systems are now being considered as effective alternatives to increase the therapeutic efficacy of the AMPs by preventing proteolysis, increase AMP accumulation at the infection sites, and reduce bystander toxicity [30]. The review paper summarizes the nano-delivery systems for AMPs, current progress, and their applications.

## 2. Antimicrobial Agents and their Activity

The correlation between microorganisms and infectious diseases is well established. Therefore, molecules that can kill, inhibit, or slow down the growth of these pathogens are vital for treatment of microbial infections. After the discovery of *Penicillium notatum* by Sir Alexander Fleming in 1928, several other antimicrobial agents were identified, and their mechanisms of antimicrobial activity have also been thoroughly investigated. Some of the antimicrobial agents and their modes of action are shown in Figure 1; they exert their antimicrobial actions by interfering with various cellular and metabolic processes of the microorganisms. Actions of antibacterial agents can either be bacteriostatic if the antimicrobial activity involves growth inhibition or bactericidal if the activity involves membrane disruption leading to death of the bacteria [31]. Sulfonamide and spectinomycin inhibit folate and protein synthesis, respectively; and are classified as bacteriostatic agents. Antibacterial agents such as vancomycin and penicillin are bactericidal agents due to their killing effect on bacteria. Other classifications of antibacterial agents are based on the mechanism of inhibition, origin, composition, and spectrum activity [32]. While these agents were initially considered highly potent against certain microorganisms, the development of microbial resistance has been reported for nearly all these antibiotics. To make matters worse, the rate of discovering new antimicrobial agents has also been declining.

The misuse and overuse of antibiotics is the main cause of AMR, and have led to the increasing resistance of human and animal pathogens to these antimicrobial agents. The mode of microbial resistance toward the antimicrobial agents include (a) alteration or modification of the drug, the drug target, and drug binding, (b) inactivation of drugs, (c) blockage or decrease in drug uptake or penetration, (d) increase in drug efflux, and (e) the degradation of the drug. In 2015, 8.7% and 24.3% AMR were reported for MRSA and *Streptococcus pneumoniae*, respectively [33], and the mechanism of bacterial resistance was through drug inactivation, increased efflux, and ribosomal protection [34].

## 3. Overview and Properties of AMPs 

AMPs, also known as host defense peptides, are short amino acid sequences or biomolecules, ranging from 12 to 100 amino acids in length. AMPs possess antimicrobial activities against various microorganisms and are crucial to both the innate and the acquired immune systems as a defense mechanism [35]. Some of these AMPs have also been shown to have antimicrobial activities against multidrug-resistant (MDR) bacterial strains. They are cationic and amphiphilic in nature [36], and these characteristics play a major role in the mechanism by which they intercalate with the phospholipid bilayer of the microbial cell membrane, leading to membrane depolarization and cell permeabilization [37]. Consequently, this will cause the release of biologically important cellular contents and subsequently result in microbial death [38,39]. 

AMPs are classified based on their sequence composition, structure, and origin; typical AMP structural conformations are highlighted in Figure 2 and Table 1. The structural and physicochemical properties of AMPs, which include charge, hydrophobicity, and amphipathicity, dictate their specificity against the target microorganisms [40].

The antimicrobial activity of these peptides appears to be more rapid when compared to conventional antimicrobial drugs [41,42], which together with new drugs are faced with the challenge of AMR. In light of this, there is a likelihood of AMPs succeeding where the conventional antimicrobial agents have failed and can possibly overcome microbial resistance due to their unique target interactions. AMPs execute their antimicrobial activities through attacks on the microbial membrane and as such would most likely require restructuring in the microbial membrane to bring about resistance to AMPs [43]. The mechanism of AMP internalization is largely dependent on the molecular properties and membrane composition. Binding and interaction occurs through the electrostatic force of attraction between the negatively charged bacterial membrane and the positively charged amino acids of the AMPs, which is followed by hydrophobic interactions between the amphipathic AMP domains and the phospholipids in the microbial membrane [41]. However, challenges such as proteolytic degradation, tissue toxicity, low stability, and difficulties associated with up-scaling must be met for AMPs to be considered as potential alternative to conventional antimicrobial drugs [44]. Modified AMPs in combination with drug delivery systems can lead to the development of novel antimicrobial agents with improved therapeutic efficacy against MDR microbes [45,46].

### 3.1. Mechanism of Action of AMPs

AMPs exert their antimicrobial activity on pathogenic microbes through different mechanism of actions. Their mechanisms are largely associated with the nature, structure, and sequence composition of the AMPs. Amphiphilicity, charge, and secondary structures of AMPs have all been associated with their different modes of action [62], suggesting that the mechanism by which the AMPs interact or disrupt the microbial membranes differ. As such, the way AMPs interact with microbial membranes is key not only to their antimicrobial action but also their application in therapeutics [63]. Other than membrane disruption, other mechanisms of action such as direct killing and immune modulation have been reported for AMPs [64]. The AMP’s ability to target the microbial membrane can occur in various ways, most notably through aggregation, barrel-stave, toroidal pore formation, and carpet model, as shown in Figure 3 [65]. All these mechanisms are extensively reviewed elsewhere [66,67].

### 3.2. Challenges of AMPs and the Role of Carriers for Improved Therapeutic Efficiency

Of all the AMPs identified, a limited number have made it to the clinical trials, and very few have been approved by the US Food and Drug Administration (FDA). Challenges such as systemic toxicity, proteolytic susceptibility, rapid clearance by the kidney, and short half-life have been mitigating against their implementation [68]. Chemical modification and delivery strategies have been suggested to overcome these challenges, and these have significantly improved their therapeutic index [29,69]. The focal point for this paper is to review the use of nano-based systems as drug carriers for AMPs.

### 3.3. Nano-Delivery Systems for AMPs

Enhancements of the properties of AMPs such as stability, toxicity, half-life, and release profile can be achieved by using delivery vehicles [70]. AMPs can be easily attached or encapsulated into delivery vehicles by covalent and non-covalent methods. This section will briefly review different vehicles or carriers for AMPs, with more emphasis on polymeric and metallic nanocarriers.

Nanotechnology-based systems through the use of nanomaterials at a size range of 1–100 nm emerged as ideal delivery systems for the AMPs. Various nanomaterials, including metallic (gold nanoparticles (AuNPs), silver nanoparticles (AgNPs), gold nanodots (Au-nanodots)) and polymeric (chitosan, poly (lactic-co-glycolic acid) or PLGA) NPs have been widely explored for the delivery of AMPs, which include but are not limited to surfactin [71], cecropin [72], and a pro-apoptotic peptide [73]. The success of nanocarriers was also demonstrated with other antimicrobial agents, which include AgNPs for ampicillin [74], silicon NPs for LL-37 [75], polymeric NPs (PLGA NPs) for colistin [76], plectasin [77], and chitosan/PGA NPs for the delivery of nisin [78], vancomycin, LL-37, cryptdin-2, and temporin B [79]. In addition, microgels, polyelectrolyte complexes, and macroscopic hydrogels have also been extensively studied for AMP delivery. These systems have been used to study the protection of antimicrobial agents against proteolytic degradation in vivo as well as to study controlled and stimuli-responsive drug release. For instance, the controlled release of Ponericin G1 (GWKDWAKKAGGWLKKKGPGMAKAALKAAMQ) was demonstrated using polyelectrolytes [80]. With the use of nanogels, the controlled release of encapsulated bradykinin in poly-2-hydroxyethyl methacrylate NPs was studied using one-pot dispersion polymerization [81]. When compared to free AMP, the encapsulated AMP showed sustained release and improved bactericidal effect against *S*. *aureus.* In another study, a hydrogel composed of nanofiber RADA16 (self-assembling peptide with the sequence: Ac-RADARADARADARADA-CONH_2_) was prepared in the presence of Tet213 (KRWWKWWRRC), and the antimicrobial activity of free RADA16 and RADA16-Tet213 was determined. In addition, the sustained release of Tet213 by RADA16 over the course of 28 days was also accounted for. At the end of the study, RADA16-Tet213 was effective against *S. aureus*. The hydrogel RADA16-Tet213 showed sustained AMP release and supported cell growth in bone mesenchymal stem cells of Sprague–Dawley rats [82]. Zetterberg et al. also investigated the use of the PEG-stabilized liposome as an AMP carrier. Protease susceptibility and the antimicrobial effects of the melittin liposome were compared to free melittin after repeated exposure to *E. coli.* Melittin liposomes demonstrated significant bactericidal activity upon second exposure when compared to free melittin, showing time-dependent release of AMP from the liposomes. In addition, melittin encapsulated within liposomes was totally protected against trypsin degradation [83]. Taken together, the results of these studies show that the use of carriers could facilitate the transition of AMPs from research laboratory into a clinical trial followed by widespread clinical use.

## 4. NPs with Antimicrobial Activity and Their Mode of Action

The search for novel antimicrobial agents has increased geometrically due to an increased incidence of microbial infections and AMR [84]. Research advancement has also led to an increased use of NPs for biomedical applications due to their broad spectrum of activities against microorganisms. In vitro studies have shown the bactericidal activity of various nanomaterials against both Gram-positive and Gram-negative bacteria [85,86,87], and similar findings were made using in vivo studies in mouse models that were infected with bacteria [88].

The antimicrobial activity of NPs has been reported for various MNPs such as zinc oxide NPs, AuNPs, and AgNPs against a number of Gram-positive and Gram-negative bacteria [89], fungi [90,91,92,93], and viruses [94,95,96]. Although most of the known polymeric NPs are commonly used as drug carriers, NPs fabricated using these polymers have been shown to possess endogenous activities. This applies to both passive (PLGA) as well as those with known antimicrobial activity (chitosan). Drugs loaded on these systems had improved biocompatibility and bio-activity. Drug-loaded PLGA-NPs exhibited higher antibacterial activity compared to the drugs alone or the unloaded NPs [97,98]. Chitosan NPs enhanced the delivery and efficacy of HIV-1 P24 protein-derived peptides [99]. Triclabendazole, which is used for the treatment of fascioliasis, is poorly soluble in water. The incorporation of this drug in chitosan NPs increased its bioavailability and stability at both low (pH 1.2) and high (pH 7.4) pH. The cytotoxic effects of the drug were significantly reduced in these nanoformulations. Thus, these nanosystems have the potential to lead to the development of orally ingested formulations for the treatment of fascioliasis [100]. This section will emphasize on MNPs with antimicrobial activity and their mechanism of inhibition.

MNPs, which include AgNPs, AuNPs, and copper oxide NPs (CuO-NPs), have all been investigated and confirmed to possess antimicrobial activity against bacteria, viruses, and fungi [101,102,103]. Although the exact mode of action of these NPs remains unclear, mechanisms such as reactive oxygen species (ROS) generation, metal ion release, and electrostatic interaction between the MNPs and the bacterial cell membrane have all been postulated. When compared to their respective salts, MNPs possess significantly higher antimicrobial activities against MDR microbes [104]. The toxicity or antimicrobial effects of AgNPs [105] and CuO-NPs [106] on microbes was size-dependent, suggesting its role in the mechanism of microbial killing. The CuO-NP size was directly proportional to their antimicrobial activity, and also small-sized monodispersed NPs showed a significant increase in antibacterial activity [106].

Specifically, the antimicrobial effect of AgNPs has been extensively studied against several microorganisms, and these are arguably some of the most promising MNPs used for the treatment of bacterial infections [107]. Characteristics such as shape, size distribution, stability, and charge make them one of the most widely explored MNPs in science, medicine, and physical science [108,109]. Figure 4 summarizes the mechanisms used in the synthesis of NPs, of which the chemical reduction method under the bottom–up approach is widely preferred over the top–down approaches. In recent years, green synthesis has been adopted to replace the toxic chemical reducing agents by benign natural resources such as microorganisms and plant extracts [110]. This method is safer and eco-friendly when compared to physical and chemical methods of synthesis. The plant-mediated synthesis is economical and make use of readily available and renewable plant materials such leaves, stems, roots, etc. Moreover, synthesis occurs in just one step, since the phytochemicals are able to act as reducing, capping, and stabilizing agents [110,111,112]. Previous studies have confirmed the safety of the biogenic method for MNP synthesis with effective antimicrobial activities against MDR bacteria [112,113].

The antimicrobial property of *Acacia rigidula* biosynthesized AgNPs was demonstrated against Gram-positive and Gram-negative MDR bacteria. The effect of the AgNPs was evaluated against *E. coli*, *P. aeruginosa*, and *Bacillus subtilis*. Their safety profile was monitored in a murine skin infection model. The outcome of this study suggested that the AgNPs are compatible for use as a therapeutic agent against infectious diseases associated with drug resistant and drug susceptible bacterial strains [98,114]. Due to their larger surface area, the surface of the MNPs can be modified to assign a specific activity by changing their surface composition. Lopez-Abarrategui et al. demonstrated that modification of citrate-coated MnFe_2_O_4_-NPs with antifungal peptide (Cm-p5) enhanced their antifungal activity in comparison to the free peptide and unmodified NPs. Cm-p5-MnFe_2_O_4_ completely inhibited *Candida albicans* (*C. albicans*) growth with a MIC of 100 µg/mL, compared to that of citrate-coated MnFe_2_O_4_-NPs at 250 µg/mL. The NPs showed no antibacterial activities against *S. aureus* and *E. coli* [115]. Therefore, MNPs could serve as excellent drug carriers due to their low cytotoxicity, ease of preparation, the ability to modify their surface, and good stability.

## 5. Nanocarriers of AMPs

NP–protein interaction possesses crucial application in biomedicine such as delivery systems and theranostic agents [116]. However, the mechanism of recognition, specificity, and selectivity are poorly understood and remain a challenge. This section discusses the nanomaterials capable of transporting AMPs into the site of action in order to overcome their afore-mentioned limitations and exact the expected therapeutic effect.

Studies have reported specific microbial resistance and their mechanisms against AMPs [117,118,119,120,121,122,123]. Pathogens can rapidly evolve and confer resistance to AMPs in vitro [124]. Resistance evolution by Baydaa et al. was arguably the first study to explore the pharmacodynamic and bacterial AMP resistance. The study showed that AMP resistance in *S. aureus* and some strains resulted not only in increased MICs but also an altered Hill coefficient (κ), resulting in steeper pharmacodynamic curves [125]. Although AMR is also observed in AMPs, MDR against AMPs is not as prevalent when compared to antibiotics [126]. Several approaches have been studied to improve the therapeutic use of AMPs. These include the combination of AMPs with traditional antibiotics since both have shown synergistic effect in the reduction of microbial resistance. Another method is the use of nanocarriers, which have been shown to reduce other side effects while exacting maximum suicidal activity against microbial populations [127].

In a bid to prevent AMR or bypass drug resistance, AMPs are now being considered as potential alternative for antibiotics. Nanomaterials, especially polymeric and MNPs, provide one of the promising drug delivery systems, as highlighted in Figure 5 [128,129,130]. Aside from the fact that some nanomaterials possess antimicrobial activities and can inhibit the growth of microbes through several mechanisms, they can also act as carriers for either antibiotics or AMPs to overcome the defense mechanisms of microbes and further enhance antimicrobial effects [131]. These materials can be functionalized with antimicrobial agents to prevent and treat microbial infections as well as to improve the effectiveness of the conventional drugs [132].

Nanomaterials such as MNPs (AuNPs, AgNPs), polymeric NPs, dendrimers, liposomes, micelles, and carbon nanotubes have all been used as carriers or transporters of drugs [133,134,135,136]. The function of these carriers is to decrease side effects, lower drug dosage, maintain constant drug levels in the blood, maximize therapeutic index, and reduce drug degradation and undesirable side effects [137]. MNPs display striking different size and shape-dependent properties when compared to their bulk material. These properties include a wide surface plasmon resonance (SPR) band, which is directly correlated to particle size, a large surface to volume ratio, biocompatibility, and low toxicity [138,139]. The MNPs’ parameters such as charge, size, and surface composition have all been reported to influence the activity of the NPs [140,141,142]. MNPs have been explored and utilized in several biomedical applications such as therapy, drug and gene delivery, probes, sensors, diagnostics, and photocatalyst.

MNPs have been shown to improve the antimicrobial activity of drugs by providing support and synergistic effects against pathogenic microbes. Specifically, MNPs have incredible physicochemical properties and have shown novel bioactivities, which can be enhanced by attaching bioactive molecules [143]. The functionalization of MNPs is achieved by conjugating different molecules through different mechanisms. MNPs employed in biological applications are usually modified with biocompatible polymers such as polyethylene glycol (PEG), proteins/peptides (e.g., bovine serum albumin), and oligonucleotides. The process of linking molecules to the surface of the MNPs can be achieved by physisorption or by taking advantage of the metal’s affinity for the sulfhydryl group of thiolated molecules. Electrostatic interactions and non-covalent conjugation can also be used to functionalize the MNPs with molecules that contain reactive groups such as hydroxyl, carboxylic group, and amine groups, which can be used to attach other biologically active molecules.

The biologically active moieties can include small drug molecules [144] or proteins [145], while targeted delivery/therapy can be achieved by conjugating target specific aptamers [146], antigen/antibodies [147], or peptides [148]. Different conjugation chemistries have been studied for effective drug loading and delivery. Examples of these chemistries include the conjugation of RR-11a endopeptidase to liposomes using the amine/carboxyl chemistry [149], the polyclonal Rabbit IgG conjugated to AuNPs using the amine/carboxylate chemistry [150], S2P peptide (CRTLTVRKC) conjugated to chitosan NPs through the amine/carboxylate, thiol/maleimide chemistry [151], liposome functionalized with adiponectin, globular domain by thiol/maleimide chemistry [152], and the J18 RNA aptamer conjugated to AuNPs by the base-pairing hybridization [153]. Immobilization of the biomolecules/AMPs on the NPs can occur in two ways; i.e., the AMPs are either temporarily (reversible) or permanently (irreversible) tethered on or within the NPs. The reversible chemistry ensures targeted delivery and release of the AMPs in its native form, and their functions are independent of the nanocarriers. The reversible nanosystems are usually responsive to biological stimuli, where a change in pH or presence of an analyte will trigger release of the attached AMPs. In the irreversible conjugation, the AMPs are permanently attached on the nanoparticle and act in synergy with the NPs [154,155,156].

Several nanoconjugates with biologically active molecules have shown promising outcomes for the treatment of some diseases. In a comprehensive study by Sibuyi et al., AuNPs were used in the development of targeted nanotherapy for cancer. The AuNPs were bifunctionalized with adipose homing (targeting) moiety and an AMP (_D_KLAKKLAK_2_/KLA) with proapoptotic activity for the selective induction of apoptosis in target cells (Figure 6). The homing peptide was conjugated to the AuNPs through an irreversible chemistry, while the KLA used a reversible chemistry where a caspase-3 cleavage site (DEVD) was used as a linker between the AuNPs and KLA. Upon internalization into the cells, the KLA is detached from the AuNPs by caspase-3 and triggers cell death through apoptosis [157]. Similarly, KLA-loaded liposomes with the adipose homing peptide accumulated in the white adipose tissues (WATs), resulting in body weight loss in obese mice models, as shown by Hossen, et al. [158]. The bifunctionalized NPs showed potential of repurposing AMPs for different applications by selectively targeted the cells that express the receptor for the targeting peptide, i.e., the colon cancer cells [157] and the endothelial cells in the WATs of obese mice [129].

### 5.1. Antimicrobial Activity of AuNPs

AuNPs, in fact most MNPs, can have dual functions by serving as both drug transporters and antimicrobial agents. Although the bactericidal effects of AuNPs were reported against MDR Gram-negative bacteria [159], AuNPs are preferably used as carriers. The therapeutic effects of AuNPs are attributed to their size, shape, and surface composition. The size and shape can be manipulated during synthesis by varying the ratios between metal precursor and reducing agents; while the surface composition can be modified by attaching biomolecules through electrostatic interaction and covalent conjugation. Targeting moieties such as antibodies are attached for target specificity purposes [160,161,162]. Anti-17β-estradiol immobilization on AuNPs achieved ultrasensitive detection of 17β-estradiol [163]. *Listeria monocytogenes* polyclonal antibodies were also conjugated to AuNPs to improve the stability of the secondary structure and the efficacy of the antibody [164].

Due to their optical properties, AuNPs can absorb light and convert it to thermal energy, and it is often used in photothermal therapy. Pedrosa et al. reported the therapeutic effect of metallic compound (TS265) functionalized AuNPs in doxorubicin-resistant cancer cells. The combination therapy resulted in 65% cancer cell death after laser treatment, and showed no cytotoxicity toward the normal cells [165]. A AuNP-based photothermal effect was also used against pathogens. Accordingly, gold nanorods (AuNRs) functionalized with either PEG (hydrophilic PEG-AuNRs) or polystyrene (hydrophobic PS-AuNRs) showed minimal antibacterial activities against *S. aureus* and *Propionibacterium acnes* (*P. acnes*) (≤85% reduction). Laser treatment enhanced the antimicrobial property of the AuNRs, resulting in ≥99.9% reduction in bacteria numbers [166]. As a result of the AuNPs’ larger surface area, multiple molecules can be attached on their surface through covalent and non-covalent conjugations. This property, together with its superior biocompatibility and targeting ability, make AuNPs a promising delivery vehicle for various biological applications. AuNPs have been used in combination with chemotherapeutic drugs such as methotrexate [167], doxorubicin [168], and oxaliplatin [169] to improve drug efficacy, delivery, stability, and to ensure a rapid increase in intracellular drug concentration [170].

The use of AuNPs, among other nanomaterials, in the regulation of the immune system by developing prophylactic and therapeutic vaccines are currently considered a promising biomedical application. Based on AuNPs’ parameters, they can be used as immunogens or as a vehicle for adjuvants and antigens [171,172] The size of AuNPs influences their uptake and internalization by cells, 3 nm AuNPs were taken up by cells through pinocytosis and were nontoxic, nonimmunogenic, and repressed ROS generation [173]. The uptake of 60 nm AuNP by cells were through endocytosis. These NPs were also nontoxic and repressed cellular responses induced by interleukin 1 beta (IL-1β) [174].

To investigate the role of drug-conjugated AuNPs, AuNPs were synthesized by the green method using sophorolipid (SL) as a reducing and stabilizing agent [175]. The antimicrobial activity of AuNPs-SL against *S*. *aureus* and *Vibrio cholerae* (*V. cholerae*) was compared to AuNPs and SL. The results indicated that SL alone exhibited antimicrobial activity against *S. aureus* but not *V. cholerae*. The AuNP-SL inhibited the growths of the microorganisms more effectively, while the AuNPs showed no activity [164]. The synergistic effect of AuNPs-SL used in combination with three antibiotics—namely, ampicillin, kanamycin, and polymyxin—were further studied against *S. aureus* and *V. cholerae*. The combined treatments had higher efficacy than the individual test agents, and the activity was highest for ampicillin followed by polymyxin. The differences in their activity could be attributed to the mechanism of the drugs, since both antibiotics interact and disrupt cell wall synthesis. Kanamycin, on the other hand, induces cell death by interfering with protein synthesis [164]. Additionally, the activity of the nanoconstructs can be channeled by attaching targeting moieties onto the NPs and achieve target specific delivery.

Incorporating antibodies that target staphylococcal protein A (aSpa) into polydopamine (PDA)-coated Au nanocages conjugated with daptomycin (Dap) improved targeting and selectivity. As shown in Figure 7, there was a complete eradication of bacteria treated with AuNP@Dap conjugates post 24 h laser exposure. Interestingly, the effects of AuNP@Dap/PDA-aSpa were immediate and persistent from 0 to 24 h, while the untargeted AuNP@Dap/PDA had a delayed effect, indicating that targeting also plays a crucial role by facilitating NP uptake and accumulation at the target site. The fact that ~50% of bacteria were viable after exposure to AuNP@PDA constructs (with and without aSpa) as a result of photothermal effects of the Au nanocages suggests that the additive effects observed with the AuNP@Dap conjugates are caused by Dap. The Au nanocages served as both carriers and photothermal agents, resulting in synergistic antibacterial effects. The AuNPs enhanced the potency of antibiotics conjugated to the nanocarriers and proved to be effective therapy against intrinsically resistant biofilm infections by MDR pathogens [176]. Therefore, the studies confirmed the effectiveness of AuNPs as nanocarriers for antimicrobial agents and that they have the potential to improve the clinical efficacy of drugs even at lower doses and prevent early drug clearance and degradation. The mechanisms by which AuNPs exert their antimicrobial activity include inhibition of transcription and energy metabolism. Simply put, AuNPs inhibit the ATPase function, which leads to ATP reduction and collapse of the membrane potential. The other mechanism is the inhibition of protein synthesis by binding to the ribosome subunit, preventing tRNA from binding [177].

### 5.2. AgNPs as Potent Antimicrobial Agents

AgNPs have been extensively studied and used commercially as antimicrobial agents. Synergistic effects have been achieved when AgNPs were used together with other antimicrobial agents. The NP conjugates are more potent and have the potential to kill MDR bacteria. The efficacy of Andersonin-Y1 against *K. pneumoniae* was improved after conjugation to AgNPs. The MIC of the AMP-tagged AgNPs was in the range of 5–15 µM when compared to the AMPs, which had a MIC of 50 µM. Further insights into their antibacterial mechanism by nuclear magnetic resonance spectroscopy and molecular dynamic simulation suggested killing by membrane pore formation through the hydrophobic collapse mechanism, thus proposing nanocarriers as potential antibiotic substitutes [130].

AgNPs have been functionalized with different molecules that have antimicrobial activity in order to achieve synergistic antimicrobial effects [178,179,180,181]. These results were further validated against *E coli*. Morphological changes, cellular uptake, and the mechanism of antimicrobial activity were determined by scanning electron microscope, transmission electron microscopes, and the lactate dehydrogenase assay (LDH), respectively. When compared to the control, the treated cells exhibited irregular shape and size, cell lysis, and cell membrane disruption, suggesting that the NPs caused the cells to rupture. In addition, the LDH assay indicated reduced cellular activities in both bacterial strains, suggesting the inhibition of the electron transport chain. This result was in agreement with the previously proposed mechanism of antimicrobial activity of nanomaterials [182].

To bypass the limitations of physical and chemical methods of MNPs synthesis, green synthesis has been explored to produce biogenic NPs that are biocompatible. Plant-mediated MNPs are synthesized using extracts from medicinal plants that are used in disease treatments. Generally, AgNPs often have more pronounced antimicrobial properties [183,184] when compared to AuNPs [185]. For example, biogenic AgNPs from *Salvia africana-lutea* (SAL) and *Sutherlandia frutescens* (SF) exhibited significant antibacterial activity against *Staphylococcus epidermidis* (*S. epidermidis*) and *P. aeruginosa*. The MIC values for SAL-AgNPs was <0.75 mg/mL. However, the MIC values of the extract was >50 mg/mL for both strains, suggesting that the NPs have enhanced antibacterial activities [183]. The activity of the biogenic MNPs can be further improved by co-treatment or biofunctionalization with other antimicrobial agents [115]. The progression of nano-based drugs into clinical trials provides evidence that nanomaterials can serve as good alternative strategies for the treatment of infectious diseases.

### 5.3. Nanohybrids for Enhanced Biocompatibility and Efficacy

The combination of two or more nanomaterials (nanohybrids) with different physico-chemical properties is becoming popular, as nanohybrids have been shown to have improved pharmacokinetics and synergistic bioactivities. Several nanohybrid compositions developed from inorganic/inorganic or inorganic/organic nanocomposites have been explored for biomedical applications. Among inorganic NPs, AgNPs are widely explored as broad-spectrum antimicrobial agents and are used in certified consumer products. In an attempt to prolong their activity and enhance their stability, inorganic/inorganic nanocomposites were created from various metal precursors. Silver has been complexed with various other metals such as copper oxide [186], gold [187], platinum [188], etc., to produce AgNP nanohybrids with synergistic or unique properties that are superior to AgNPs alone. The nanohybrids improved the efficacy of antimicrobial agents [189] when used as carriers or in combination therapy. In an interesting study by Kazemzadeh-Narbat et al., titanium oxide nanotube implants consisting of multiple layers of thin films were synthesized and used for the time-dependent release of an AMP (HHC-36/KRWWKWWRR-NH_2_). This was done by encapsulating the AMP with calcium phosphate and titania nanotube. The study demonstrated the therapeutic efficacy of the nanotubes against *S. aureus* and *P. aeruginosa* through the controlled release of AMP [190].

Similarly, the inorganic/organic nanocomposites present a benign strategy that can be used to encapsulate the inorganic nanomaterial within the organic nanomaterial, which may render the nanocomposite more bio-friendly, increasing its bioavailability and safety profiles. Liposomes and chitosan NPs are some of the bioinert and biodegradable nanomaterials used to avoid the proteolytic degradation of drugs. Moorcroft et al. created AuNR/AMP-liposome loaded hydrogels that are responsive to laser irradiation due to the presence of AuNRs. The study demonstrates the controlled release of AMP liposomes in response to laser irradiation (Figure 8a). IK8 (IRIKIRIK-CONH_2)_, which is the AMP used in this study, was shown to be potent against the test microorganisms (*P*. *aeruginosa* and *S*. *aureus*). IK8 is susceptible to proteolytic enzyme degradation, leading to loss of the antimicrobial activity. However, the entrapment of IK8 inside liposomes protected the AMP from trypsin-induced degradation for up to 5 h (Figure 8b), and further encapsulation with AuNR in hydrogel allowed for the stimuli-responsive release of the AMPs. Thus, the AuNR/AMP liposome-loaded hydrogel hybrids enhanced the AMP stability, protected it from degradation, facilitated the controlled release of AMP, and facilitated the retention of the nanomaterials within the hydrogel. The AuNR/AMP liposome-loaded hydrogel showed dual and synergistic effects, resulting from the antibacterial effects of IK8 and the photothermal activity of the AuNRs. A single dose was used in two cycles of treatment [191].

In an independent study, a similar strategy was used to develop pH-sensitive AuNP-liposome-loaded hydrogels. Carboxyl-modified AuNPs (AuC) were used to stabilize cationic liposomes (AuC-liposomes), which were then loaded into a polyacrylamide hydrogel, and the AuC-liposomes release profile was studied in *S. aureus* at pH 7.4 and 4.5. Higher uptake was observed at pH 4.5, suggesting that AUC-liposomes are possibly released from the hydrogels at pH 7.4 and the AuC detaches from the liposomes at acidic pH below the pKa of the carboxylic group. This proved that these systems can be used to deliver antimicrobial agents for the treatment of bacterial infections. In vivo studies showed that the formulation was also biocompatible and showed no toxicity when topically administered daily on a skin of mice for 7 days [192].

## 6. Nanocarriers in Clinical Trials

Several functional nanomaterials that has been investigated in preclinical and clinical studies led to the development of nanomedicines that are currently on the market, most notably some clinically approved liposome drug formulations and metallic imaging agents [193]. Most of the nanocarriers in preclinical and clinical trials as well in clinical use are for cancer targeting [194]. Lipid and polymeric nanocarriers are the most widely used, and examples of these carriers in clinical trials include LiPlaCis, a lipid-based nano-formulation with cisplatin currently in phase II for refractory solid tumors and NK105 conjugated with paclitaxel currently in phase II for the treatment of gastric cancer [195]. Moreover, the utilization of nanotechnology-based treatments for diseases other than cancer has increased exponentially in recent years. Antimicrobial therapy is another pivotal clinical focus that is being investigated for the advancement of nanomedicine. MNPs can serve as carriers and also as potential antimicrobial agents. Specifically, AgNPs can diffuse through the microbial cell membrane, leading to toxicity through Ag^+^ release. This nanomaterial has been extensively studied and used in medicine for decades. Due to their activities, AgNPs with antimicrobial properties were approved by the FDA for wound therapy [196,197]. Currently, there are nine ongoing clinical trials investigating AgNPs as an antibacterial therapy in different diseases.

AMPs have been ascertained as outstanding alternative antimicrobial agents to overcome AMR, and a growing body of evidence suggests that AMPs are swiftly gaining more attention for their potential clinical application as they present remarkable benefits over conventional antibiotics. AMPs are uniquely implicated in all life forms and display significant roles in the innate immune system. Some AMPs are evolutionarily conserved, which could be an indication that these molecules may be capable of restricting tendencies for the development of microbial resistance.

Despite the therapeutic benefits of AMPs, only a few AMP-based formulations have successfully progressed into clinical trials. [198]. AMPs in clinical trial are grouped based on their mechanism of action on cell membrane of microbes (Temporin10a [199], Ruminococcin C [200]), immune system (IDR-1002 [201]), and intracellular functions (HB-107 [202], Buforin II [203]. Specific modifications to AMPs can be used to improve their delivery, biological activity, and stability as well as reduce toxicity. Advances in nanotechnology for drug delivery have been extensively reviewed. The successful application of nanotechnology to develop improved drug delivery systems has been demonstrated successfully, and therefore, its application for the delivery of AMPs is also attainable and can lead to the development of novel antimicrobial agents that can fight MDR.

It is further crucial to investigate the fundamental biological effect, biodistribution, and pharmacokinetics of MNPs, most especially silver and other nano-based nanocarrier for clinical applications.

### Merits and Limitations of Nanocarriers

As described above, NPs have been involved in different applications in the field of biomedicine and have proven to be effective drug delivery vehicles and a potential alternative antimicrobial agent. Nanomaterials such as MNPs, liposomes, dendrimers, polymeric, and carbon nanotubes have been widely implicated in the design of AMPs with enhanced activity toward MDR microorganisms. However, drawbacks such as cytotoxicity, conjugation protocols, stability profiles, and shelf-life have been reported for the AMPs. Table 2 shows some of the nanocarriers that were used as vehicles for AMPs, together with their limitations. Carbon nanotube synthesis is costly, while these NPs have poor solubility. Liposomes have low loading capacity and could also induce immune response. While liposomes are biodegradable and both hydrophobic and hydrophilic drugs can be loaded, factors such as drug-loading efficiency and immunogenicity remain a challenge. Dendrimers are monodispersed molecules with a high control over the critical molecular design parameter. However, the cost of synthesis together with its non-specificity remain major limiting factors for dendrimers. Studies have reported the biocompatibility and nature-dependent biodegradability of polymeric NPs. This class of carriers is easy to modify and to control drug release. Its disadvantages include low cell affinity and the toxicity of their by-products.

## 7. Conclusions and Future Perspectives

As a challenge, microbes evolve geometrically faster than the discovery and implementation of antibiotics. Moreover, these antibiotics gradually lost their antimicrobial activity and drug-resistant bacteria appeared with antibiotics overuse and even the misuse of several other antibiotics. Although, AMPs have been thought as an alternative to antibiotics, there is still a pool of antibiotics in AMPs undiscovered and poor pharmacokinetics of peptide drugs is a disadvantage to the application of existing AMPs.

The use of AMP delivery systems can facilitate their progress into clinical trials and ultimately be fundamental for their implementation.

Antibiotics exert their action by targeting specific cellular components or metabolic intermediates, resulting in microbial growth inhibition or microbial death. The most common mechanism is through the inhibition of vital enzymes involved in microbial growth and metabolism. This could result in genetic mutations, leading to microbial drug resistance. Conversely, AMPs due to their amphipathic properties bind to the membrane bilayer of the negatively charged bacteria, which allows rapid penetration. This process is not affected by mutations. More so, AMPs carry out their antimicrobial activity on the entire cellular membrane, making it burdensome for microbes to develop resistance against them. AMPs capable of immune regulation make their mode of action comprehensive. Antibiotics undergo detoxification, and their complete renal clearance is not guaranteed. However, the metabolic degradation of AMPs involves monomeric amino acids that can be channeled to essential biosynthetic pathways.

Due to the continuous resistance development against current antibiotics and antimicrobial agents, novel therapeutics are urgently needed. AMPs have emerged as alternatives due to the mechanism by which they cause physical disruption of the phospholipid bilayer of the pathogens, resulting in their death. The metabolic requirement of membrane repair limits the risk of drug resistance against these peptides. Nonetheless, instability and proteolytic degradation among others have been correlated with AMPs and have in turn limited their implementation. Drug delivery systems are proposed as the right approach toward their uptake, release (sustained, controlled, and triggered), and protection against proteases in order to overcome MDR challenges. Nanomaterials are now being used as drug delivery systems to improve therapeutic activity and reduce undesirable side effects. With the broad therapeutic involvement of NPs, it is worth establishing the mechanism by which their conjugation, functionalization, encapsulation, and complexes can influence bacterial population. Nanocarriers such as MNPs, polymeric, and liposomes among others have all been reported as effective drug carriers with good therapeutic indexes.

Of specific benefit, NPs are able to target infection sites. As such, the synergistic activity of the nanocarriers together with AMPs in cell wall penetration, particle aggregation, ROS formation, and inhibition of cellular activities are extremely important in fighting pathogens and MDR infections. Although challenges of AMP-carrier systems include finding an appropriate carrier, entrapment efficiency, and conjugation chemistry; there are numerous ongoing research studies for AMP-nanocarriers optimization. When compared with the number of AMP complexed with nanocarriers in clinical trials, basic studies with in vitro end points are geometrically greater. With ever-expanding AMP discovery, more in vivo studies are required to understand the physiological barriers and immunological responses in order to simplify the challenges in clinical trials. Nanotechnology is capable of revolutionizing the world of medicine, and AMP nanocarriers are worth the investment in order to tackle MDR pathogens.

## Figures and Tables

**Figure 1 pharmaceutics-13-01795-f001:**
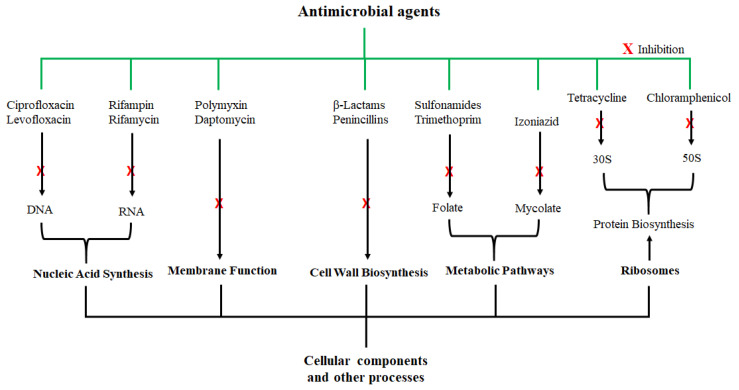
Examples of antimicrobial agents and their modes of action. These compounds are classified according to their cellular or molecular targets.

**Figure 2 pharmaceutics-13-01795-f002:**
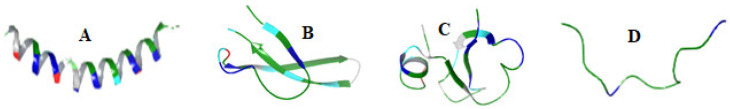
Structural conformations of AMPs. (**A**) α-helical, (**B**) β-sheet, (**C**) αβ-peptides, and (**D**). Non-αβ peptides or extended structure. Images were deciphered by Schrodinger software v2020-3 after structural retrieval from the Protein Data Bank (PDB) at https://www.rcsb.org/ (accessed on 20 May 2021) using the PDB IDs: 2K6O, 1ZMM, 1FD3, and 1G89 for A to D, respectively. AMPs are colored by properties.

**Figure 3 pharmaceutics-13-01795-f003:**
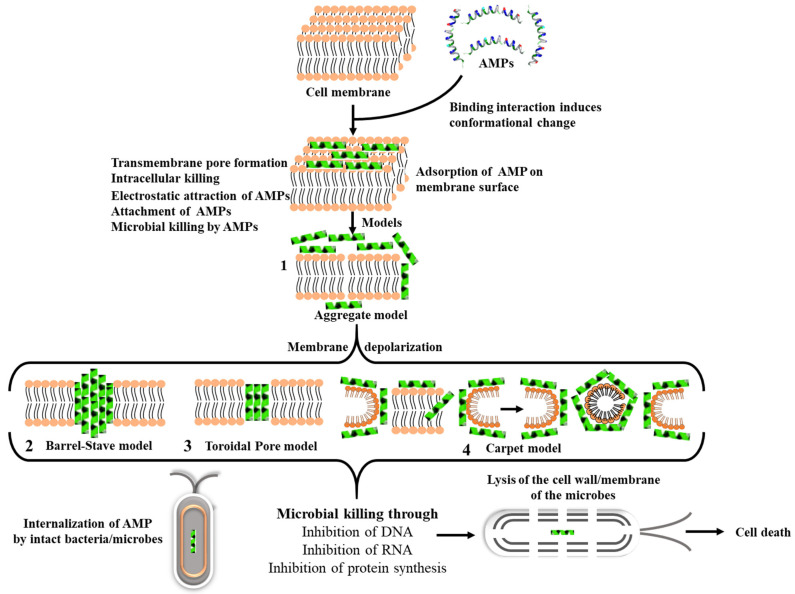
Mode of action used by AMPs to target and disrupt the bacterial membrane, leading to cell lysis and bacterial death.

**Figure 4 pharmaceutics-13-01795-f004:**
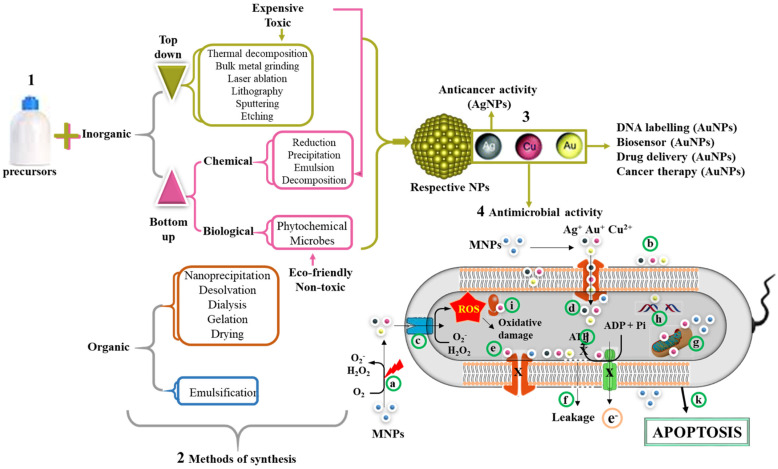
Synthesis of organic and inorganic NPs and their antimicrobial mechanism. MNPs can be synthesized using either a bottom–up or top–down approach (1). The methods of NP synthesis are broadly classified into physical, chemical, and biological methods (2). The reduction of metallic salts by these methods leads to the formation of MNPs. Examples of MNPs include AgNPs, AuNPs, and CuONPs (3). Various applications of MNPs include DNA labeling, biosensor, drug delivery, anticancer, and antimicrobial properties (4). The possible mechanism of their antimicrobial activity involves release of metallic ion when excited by laser (a) or in the presence of oxygen. The size of MNPs exhibited electronic effects and thus improved surface attraction (b). These metallic radicals in addition to their penetrative ability can easily pass through membrane channels (c). The MNPs triggers the generation of ROS inside the cells, which in turn leads to cellular damage. In addition, MNPs inhibit the channel transport of solutes/ions (e). The accumulation of these metallic ions leads to membrane depolarization and ultimately membrane leakage of cellular contents (f), mitochondrial dysfunction (g), DNA damage (h), ribosome disassembly (i), and inhibition of the electron transport chain and ATP synthesis (j). Collectively, all these processes can lead to cell death through apoptosis (k).

**Figure 5 pharmaceutics-13-01795-f005:**
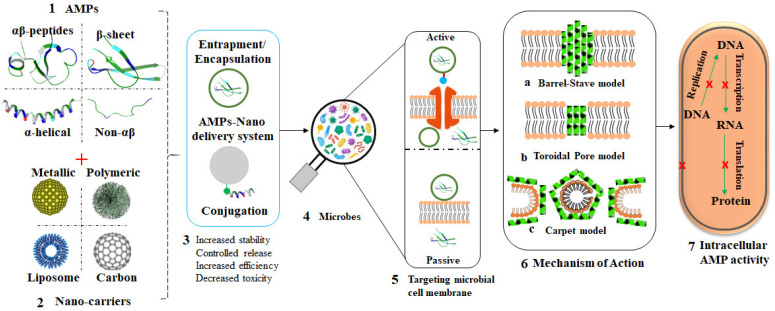
Overview of AMP-loaded nanocarriers and their mode of action. Structurally, AMPs are classified into four groups (1); different nanocarriers have been studied as effective carries of AMPs (2); different AMP nanoformulations can be obtained through various chemistries between peptides and nanocarriers (3); exposure of microbes to these AMP nanoformulations (4); via passive or active transportation (5); leading to bacterial membrane attack by the AMPs through various AMP-dependent mechanisms (6) and ultimately bacterial death (7).

**Figure 6 pharmaceutics-13-01795-f006:**
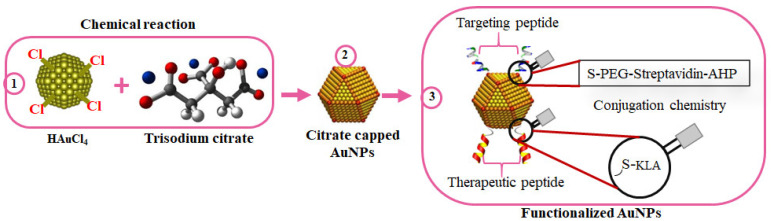
Surface chemistry of bifunctionalized citrate-capped AuNPs. AuNPs were chemically synthesized using trisodium citrate as the reducing agent (1); to form citrate-capped AuNPs (2); while exploring the AuNP-thiol affinity for the attachment of the targeting and therapeutic peptides (3).

**Figure 7 pharmaceutics-13-01795-f007:**
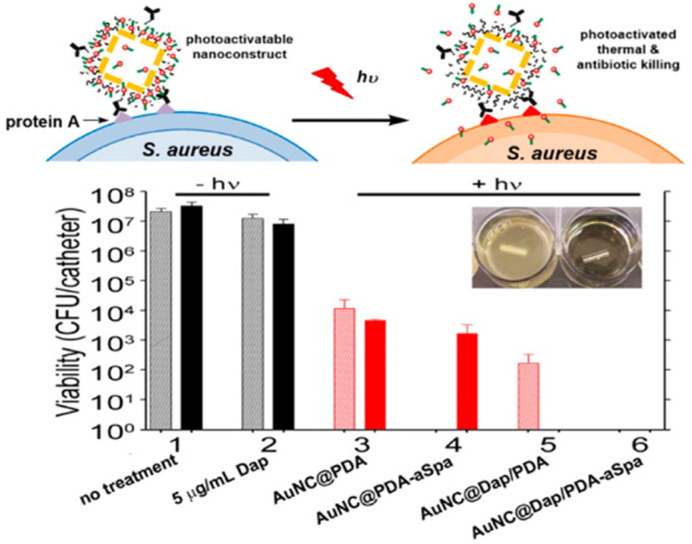
Synergistic effect of laser-induced and controlled release of Dap in *S*. *aureus*. Reprinted with permission from Copyright © 2016 American Chemical Society: https://pubs.acs.org/doi/full/10.1021/acsinfecdis.5b00117, accessed on 20 May 2021, further permissions related to the material excerpted should be directed to the ACS.

**Figure 8 pharmaceutics-13-01795-f008:**
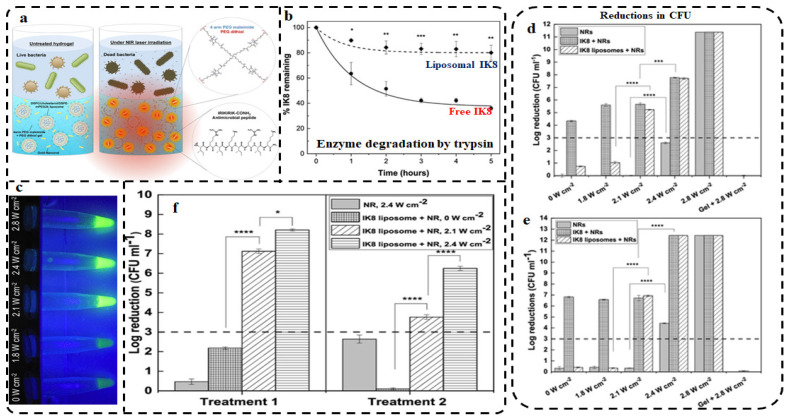
Controlled release and thermal enhancement of IK8-conjugated AuNR-loaded hydrogel. Mechanism of IK8 release triggered by laser irradiation of AuNRs (**a**); proteolytic effect of trypsin on IK8 and liposome IK8 over the course of 5 h, * *p* ≤ 0.05, ** *p* ≤ 0.01, *** *p* ≤ 0.001 (**b**); irradiation of the AuNR-loaded hydrogels at 850 nm with ranging intensities from 0 to 2.8 W/cm^2^ (**c**); Antibacterial effect of laser-irradiated hydrogels against *S. aureus*, *** *p* ≤ 0.001, **** *p* ≤ 0.0001 (**d**); and *P. aeruginosa*, **** *p* ≤ 0.0001 (**e**); and activity of different formulations on *S. aureus* over two treatment cycles after 5 (treatment 1) and 10 (treatment 2) min laser irradiation, * *p* ≤ 0.05, **** *p* ≤ 0.0001 (**f**). Reprinted with permission: Copyright © 2020 American Chemical Society: https://pubs.acs.org/doi/10.1021/acsami.9b22587, accessed on 20 May 2021, further permissions related to the material excerpted should be directed to the ACS [191].

**Table 1 pharmaceutics-13-01795-t001:** Classifications of some AMPs with specific examples.

Groups	Characteristics	Examples	Mode of Action	Refs
α-helical peptides	Amidated C-terminus, N-terminal signal peptides	FALL-39 Magainins Cecropins	Pore formation	[47]
				[48,49]
				[50]
β-sheet	cationic with disulfide bridges	β-defensins	Membrane disruption	[51,52]
		plectasin		[53]
		protegrins		[54]
Extended AMPs or Non-αβ peptides	Contains proline, arginine, tryptophan, glycine or histidine rich amino acids	Indolicidin	Membrane disruption Disruption of intracellular function	[55]
		Bactenecins		[56]
		Histatins		[57]
Loop peptides		Dodecapeptides Tachyplesins Protigrin-1 Bactenecin-1 Ranalexin Brevinin 1E Lactoferricin	Disruption of bacterial membrane	[58] [59,60] [61]

**Table 2 pharmaceutics-13-01795-t002:** Advantages and limitations of various nanocarriers.

Nanocarriers	Advantages	Limitations
MNPs	Multipurpose High surface to volume ratio	Cytotoxicity Shelf-life Solubility
Liposomes	Biodegradable Hydrophobic and hydrophilic molecules can be loaded	Loading efficiency Immunogenicity
Dendrimers	High control over the critical molecular design parameter	High cost of synthesis Non-specific toxicity
Carbon nanotubes	Soluble in water Multiple application	High cost of synthesis Less degradable
Polymeric NPs	Easy modification Biocompatibility Nature-dependent biodegradability Time-dependent drug release.	Low cell affinity toxicity of byproducts.

## Data Availability

Not applicable.

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
