# Peer review of "Nanotechnology-Based Delivery Systems for Antimicrobial Peptides"

_pharmaceutics, 2021, doi:10.3390/pharmaceutics13111795_

Round 1
Reviewer 1 Report
Well-designed review work. Interesting and rich drawings. A large amount of valuable literature.
I recommend publishing this manuscript in Pharmaceutics.
Author Response
Reviewer 1
Comments and Suggestions for Authors
Well-designed review work. Interesting and rich drawings. A large amount of valuable literature. I recommend publishing this manuscript in Pharmaceutics.
Response
Thank you for the good work and comments. We (the authors) are indeed grateful.
Reviewer 2 Report
The review received for evaluation is titled 'Nanotechnology-Based Delivery Systems...' and has as correspondence author M. Meyer. The manuscript has 183 references written in the correct format, however, only for some of them there is included the DOI number. The Abstract provide an overview of the work. The work contains 8 figures and 2 tables. Pictures inserted are representative and easy to be understood, getting in a single point the vision of the review. The conclusion are also representative.
There is a difference in writing subchapters, for example 1.1 is written in bold, while 2.1 in italics- consistence throughout the manuscript is necessary.
All together, the manuscript is very well written and organized and it is supposed to be of high interest for readers; antimicrobial peptides is indeed a subject of top science, from both experimental and applicative point of view. Therefore, I recommend the work for publication as it is.
Author Response
Comments and Suggestions for Authors
There is a difference in writing subchapters, for example 1.1 is written in bold, while 2.1 in italics- consistence throughout the manuscript is necessary.
Response
Thank you for the comment, we have ensured uniformity of the sections and the subsections of the manuscript. We have also done English editing from the beginning to the end of the manuscript.
Reviewer 3 Report
This is a well-written and well-organized review of nanotechnology-based delivery systems for AMPs. It provides a comprehensive view of the field and will be of interest to a large number of researchers in the area. There are a couple of minor issues that should be addressed and one major issue. These are outlined below.
Major issue: In describing the interactions of AMPs with nanoparticles, the authors begin (Section 2.3) with a discussion of several peptides associated with various types of nanoparticles. However, it is not until Page 10 (and Figure 5) that the authors include discussion of how peptides are associated with nanoparticles. This discussion is essential to the understanding of how the AMPs interact with bacteria and host tissues. The authors discuss encapsulation and conjugation (conjugation should be broken down into reversible and irreversible). Encapsulation should deliver the AMP in a monomeric, unmodified form; reversible conjugation should do the same; while irreversible conjugation leaves the AMP on the nanoparticle to act with direct synergy. These considerations color the results of each experiment and should be discussed early in the review and subsequent examples should include descriptions of these connections between AMPs and nanoparticles.
Minor issues:
- Line 36: replace “dose” with “use”
- Line 69: remove the words “such as”
- Lines 74 and 75: correct “P2X7”
- Line 118: remove the word “great”
- Line 182: capitalize FDA
- Line 192: delete the word “on”
- Table 2: change “ration” to “ratio”
Author Response
Reviewer 3
Comments and Suggestions for Authors
Major issue:
In describing the interactions of AMPs with nanoparticles, the authors begin (Section 2.3) with a discussion of several peptides associated with various types of nanoparticles.
However, it is not until Page 10 (and Figure 5) that the authors include discussion of how peptides are associated with nanoparticles. This discussion is essential to the understanding of how the AMPs interact with bacteria and host tissues.
The authors discuss encapsulation and conjugation
(conjugation should be broken down into reversible and irreversible).
Encapsulation should deliver the AMP in a monomeric, unmodified form; reversible conjugation should do the same;
while irreversible conjugation leaves the AMP on the nanoparticle to act with direct synergy.
These considerations color the results of each experiment and should be discussed early in the review and subsequent examples should include descriptions of these connections between AMPs and nanoparticles.
Response to major issues
The authors have briefly described the different immobilization methods in line 408, to high the differences between the two systems.
Minor issues:
- Line 36: replace “dose” with “use”
- Line 69: remove the words “such as”
- Lines 74 and 75: correct “P2X7”
- Line 118: remove the word “great”
- Line 182: capitalize FDA
- Line 192: delete the word “on”
- Table 2: change “ration” to “ratio”
Response to minor issues
All the minor issues have been fixed as recommended. We have also done English editing from the beginning to the end of the manuscript.
Reviewer 4 Report
The manuscript entitled “Nanotechnology-Based Delivery Systems for Antimicrobial
Peptides” by Fadaka et al talks about the identification of risk factors for antimicrobial resistance (AR) and how the conventional antibiotics are failing to treat diseases. In fact the World Health Organization ranked antibiotic resistance as a priority disease. Antimicrobial peptides have come up as key molecules to fight antimicrobial resistance.
Overall, the study is clear and concise. The introduction is relevant and theory based. Sufficient information about the present study rationale and procedures are provided for the readers. The language and diagrams are generally appropriate, although clarification of a few details are required. Overall, the results are clear and compelling. The authors make a systematic contribution to the research literature in this area of investigation particularly when the antimicrobial resistance is creeping in the antimicrobial peptides too. The NP formulations might help to fight it.
Specific comments below
- There is abundant literature in the file of antimicrobial resistance. I am glad they mentioned MRSA. However, the authors should talk about ESKAPE pathogens too. (Enterococcus faecium, Staphylococcus aureus, Klebsiella pneumoniae, Acinetobacter baumannii, Pseudomonas aeruginosa, and Enterobacter species). They may put it in the introduction.
- Unfortunately, antimicrobial resistance is also creeping in antimicrobial peptides too. Can the authors talk about the role of nanoparticles to prevent it?
- Please put a hyphen between the Gram Positive Bacteria. Instead of writing Gram positive bacteria, write Gram-positive bacteria. Do the same for Gram-negative bacteria as well.
- Can the authors provide the ATCC numbers of the microbial strains they mentioned in their study?
Author Response
Reviewer 4
Comments and Suggestions for Authors
- There is abundant literature in the file of antimicrobial resistance. I am glad they mentioned MRSA. However, the authors should talk about ESKAPE pathogens too. (Enterococcus faecium, Staphylococcus aureus, Klebsiella pneumoniae, Acinetobacter baumannii, Pseudomonas aeruginosa, and Enterobacterspecies). They may put it in the introduction.
Response
We have updated the introduction section with “The emergence of microorganisms with acquired multi-drug resistance is one of the major concerns to date [1]. Enterococcus faecium, Staphylococcus aureus, Klebsiella pneumoniae, Acinetobacter baumannii, Pseudomonas aeruginosa, and Enterobacter species (ESKAPE pathogens) characterized by potential drug resistance mechanisms are associated with nosocomial infection and therefore, place a significant burden on the healthcare systems and global economic costs [2,3]. Efforts to impede the spread of these pathogens have been hindered by their ability to resist antibacterial drugs.” As instructed.
- Unfortunately, antimicrobial resistance is also creeping in antimicrobial peptides too. Can the authors talk about the role of nanoparticles to prevent it?
Response
This is crucial to the importance of this review. We have therefore added information about the resistance to AMPs.
“Studies have reported specific microbial resistance and their mechanisms against AMPs [1-7]. Pathogens can rapidly evolve and confer resistance to AMPs in vitro [8]. Resistance Evolution by Baydaa and co was arguably the first study of explore the co-evolution of pharmacodynamic and bacterial AMP resistance. The study showed that AMP resistance evolution in S. aureus and some strains resulted not only in increased MICs, but also alter Hill coefficient (κ), resulting in steeper pharmacodynamic curves [9]. Although antimicrobial resistance is beginning to creep in AMPs for their intended use, MDR against AMPs is not as prevalent when compared to antibiotics [10]. Several approaches have therefore been studied to improve the therapeutic use of AMPs. These include the combination with traditional antibiotics since both have shown synergistic effect in the reduction of microbial resistance. Another method in the use of nanocarriers which have been shown to reduce other side effects while exacting maximum suicidal activity against microbial populations [11].”
- Please put a hyphen between the Gram Positive Bacteria. Instead of writing Gram positive bacteria, write Gram-positive bacteria. Do the same for Gram-negative bacteria as well.
Response
We have effected this change as recommended.
- Can the authors provide the ATCC numbers of the microbial strains they mentioned in their study?
Response
Authors would have loved to include the ATCC number of the mentioned strains but virtually all the research papers reviewed did not provide this number. Adding the ATCC of the ones that provided it will deflect this manuscript from ensuring uniformity. As such, we would like to suggest that the strains should be without ATCC numbers. In addition, readers can follow the references to the main paper if need be.
We have also done English editing from the beginning to the end of the manuscript.